# Primary Cutaneous Nocardiosis (Lymphangitic Type) in an Immunocompetent Patient: A Case Report

**DOI:** 10.3390/microorganisms13051022

**Published:** 2025-04-29

**Authors:** Hilayali Aguilar-Molina, Sonia Toussaint-Caire, Roberto Arenas, Juan Xicohtencatl-Cortes, Luary C. Martínez-Chavarría, Rigoberto Hernández-Castro, Carmen Rodriguez-Cerdeira

**Affiliations:** 1Servicio de Dermatología, Hospital General “Dr. Manuel Gea González”, Tlalpan, Ciudad de Mexico 14080, Mexico; ham87e@gmail.com; 2Servicio de Dermatopatología, Hospital General “Dr. Manuel Gea González”, Tlalpan, Ciudad de Mexico 14080, Mexico; tussita@hotmail.com; 3Sección de Micología, Hospital General “Dr. Manuel Gea González”, Tlalpan, Ciudad de Mexico 14080, Mexico; rarenas97@hotmail.com; 4Fundación Vithas, Grupo Hospitalario Vithas, 28043 Madrid, Spain; 5Laboratorio de Bacteriología Intestinal, Hospital Infantil de Mexico Dr. Federico Gómez, Cuauhtémoc, Ciudad de Mexico 06720, Mexico; juanxico@yahoo.com; 6Departamento de Patología, Facultad de Medicina Veterinaria y Zootecnia, Universidad Nacional Autónoma de Mexico, Coyoacán, Ciudad de Mexico 04510, Mexico; luary@unam.mx; 7Departamento de Ecología de Agentes Patógenos, Hospital General “Dr. Manuel Gea González”, Tlalpan, Ciudad de Mexico 14080, Mexico; 8Dermatology Department, Grupo Hospitalario (CMQ Concheiro), Manuel Olivié 11, 36203 Vigo, Spain; 9Department of Health Sciences, University of Vigo, Campus of Vigo, As Lagoas, 36310 Vigo, Spain; 10European Women’s Dermatologic and Venereologic Society (EWDVS), 36700 Tui, Spain

**Keywords:** nocardiosis, *Nocardia brasiliensis*, lymphangitic type, immunocompetent patient, 16S ribosomal rDNA, treatment

## Abstract

Cutaneous nocardiosis is an uncommon bacterial infection caused by *Nocardia* spp.; *Nocardia brasiliensis* is the agent involved in most cases. This infection is acquired through the direct traumatic inoculation of soil, plants, or other substrates where the bacteria are found. Clinically, it usually manifests as an erythematous ulcerated nodule. In one-third of cases, nodules or gummas are distributed over the lymphatic pathways that resemble lymphocutaneous sporotrichosis. Its manifestations vary and can present acutely or more frequently with a latent clinical picture over time. Diagnosis is established mainly by Gram staining, biopsy, exudate culture, and molecular biology. *Nocardia* infections can recur, implying that antimicrobial therapy must be prolonged (between 6 and 12 months) and involve monitoring patients for at least 6 months after the end of treatment. Early diagnosis and targeted treatment may reduce patient mortality rates. We report the case of an 82-year-old woman who presented with four nodules with a lymphangitic spread on her left hand and forearm, one week after the trauma. Molecular identification was performed using 16S rDNA gene sequencing, and *Nocardia brasiliensis* was identified.

## 1. Introduction

Nocardiosis is a rare opportunistic infectious disease caused by bacteria of the genus *Nocardia* belonging to the *Actinomycetales* family [1]. Acquisition occurs through direct inhalation of contaminated particles or direct inoculation through the skin [2]. Skin infections can be caused by direct inoculation (primary) or haematogenous or lymphatic dissemination, mainly in immunocompromised hosts (secondary infections) [3]. Haematogenous dissemination frequently occurs in the central nervous system, retina, heart, joints, and kidneys [4,5]. Infection is not transmitted from person to person [5]. However, its prevalence remains unclear; reports warn of an increase in the number of cases. This may be related to the prevalence of immunocompromised patients and the improvements in diagnostic techniques [4].

The appearance of skin nocardiosis is not specific; therefore, a skin biopsy should be performed to identify *Nocardia* or co-pathogens. *Nocardia* infections are most frequently associated with bacteraemia [6], which may indicate either a disseminated infection or intravascular device-related bacteraemia. Clinical manifestations of patients are nonspecific; it often causes multiple organ involvement, and early diagnosis and prompt aggressive interventions are important to improve the outcome of this disease [7].

Primary skin nocardiosis is more frequent among immunocompetent individuals and may present as a mycetoma, superficial skin infection, or lymphocutaneous infection. The infection can spread to the regional lymph nodes and produce a single or linear chain of nodular lesions, which typically present as an ulcerated draining lesion with lymphangitic spread and overlying subcutaneous nodules [8]. Thus, any cutaneous lesion in an immunocompromised patient should be suspected as nocardiosis. Superficial abscesses or localised cellulitis can develop after skin inoculation. Cutaneous nocardiosis can resemble soft tissue infections produced by *Staphylococcus aureus* or *Streptococcus* spp.; however, this form of nocardial disease is usually more indolent [9]. Skin and soft tissue nocardiosis can have various appearances, mimicking any type of skin or soft tissue infection [10]. The cutaneous form can also be characterised by pustules, abscesses, blisters, ulcers, and/or cellulitis at the site of inoculation or the lymphangitic form [11]. The clinical presentation is acute, rapidly progressive, and highly inflammatory [11,12,13]. Multifocal lesions usually represent dissemination and are more prevalent in immunocompromised individuals. Such lesions can appear as papules, nodules (sometimes ulcerated), superficial skin infiltrates, or subcutaneous or muscular abscesses [14].

*Nocardia* is a Gram-positive, aerobic, filamentous, branched, fungus-like, and slow-growing bacterium that can grow in media for bacteria, fungi, or mycobacteria [1,2,4,6,15]. Typically, the colonies are chalky white in appearance and produce a wet, earthy odour [4]. Other properties that can help identify this species in the laboratory include mild acid-alcohol resistance, resistance to lysozyme, and hydrolysis of casein, tyrosine, xanthines, and hypoxanthines. Reference laboratories have developed molecular biology techniques (Polymerase chain reaction) or incorporated mass spectrometry for diagnosis [4,16,17]. To date, no serological methods are available for identifying *Nocardia* spp. [2,9].

Primary cutaneous nocardiosis is a rare infection due to partially acid-fast aerobic actinomycetes of the genus *Nocardia*, which accounts for 5–7% of total infections by *Nocardia*. *N*. *brasiliensis* (80%), *N*. *asteroides*, and *N*. *africana* are the most frequently isolated agents in cutaneous nocardiosis, with *N. transvalensis* being rarely isolated [18,19]. *Nocardia brasiliensis* is most frequently associated with cutaneous diseases and is mainly isolated from areas with tropical or subtropical climates [17]. This microorganism is a saprophytic bacterium that is widely distributed in nature, predominantly found in the soil, dust, compost, sand, decaying vegetation, and aquatic environments [17].

Here, we report a case of cutaneous nocardiosis in an immunocompetent 82-year-old woman who presented four nodules with a lymphangitic spread on her left hand and forearm, a ten-day-old injury, and a history of frequent agricultural work. Bacteriological culture and molecular identification based on the sequencing of the 16S rRNA subunit allowed the diagnosis of *Nocardia brasiliensis*. The treatment was based on a high dose of oral amoxicillin–clavulanic acid, with complete resolution of clinical manifestations after three months of treatment. This clinical case suggests that a high dose of amoxicillin–clavulanic acid is an efficacious monotherapy for acute primary cutaneous nocardiosis.

## 2. Case Report

An 82-year-old female resident of Mexico City, Mexico, presented to the Dermatology Department at “Dr. Manuel Gea Gonzalez” General Hospital. Her general physical examination revealed four erythematous nodules with central ulceration affecting the left thumb, wrist, and inner side of the left forearm, with lymphatic spread and mild pain in the region. Some lesions were surrounded by extensive zonal erythema and seropurulent and bloody discharge, with scattered crusting and scarring. The patient reported a previous inoculation with a cactus’s thorn one or two weeks prior (Figure 1A).

On admission, the patient had no systemic symptoms, such as fever, chills, fatigue, dizziness, or headache, and she was haemodynamically stable. The laboratory results were within normal limits, and no leukocytosis or elevation of inflammatory markers was observed. The laboratory values were as follows: white blood cell count (WBC) of 10.4 cells/mm^3^ (reference, 4000–10,000), haemoglobin 14.0°g/dL, haematocrit 40%, platelets, 24.2 × 10^4^/mm^3^, normal sedimentation rate of 15 mm/h (reference range: 0–22 mm/h), and C-reactive protein of 0.9 mg/L (reference range: 0–8 mg/L).

Biopsies, cultures, and smears were performed. Histopathological examination showed suppurative and granulomatous infiltrates (Figure 2); however, no bacterial or fungal structures were observed on Gram, Gomori–Grocott, or PAS staining. Bacteriological culture was performed in Sabouraud agar incubated at 37 °C with a 5% CO_2_ atmosphere. A white folded colony, with a dry surface and hard consistency with a “popcorn” aspect, was observed after four days of incubation. Gram staining revealed the presence of weak Gram-positive, branched filamentous bacilli, which retained acid-fast staining with the Kinyoun method. Sensitivity tests were not performed.

Genomic DNA was isolated from pure cultures using the DNeasy Blood and Tissue Kit (Qiagen, Ventura, CA, USA) according to the manufacturer’s instructions. The molecular identification was performed by 16S ribosomal rDNA gene amplification using a set of primers (5′-GGATCCTTTTGATCCTGGCTCAGGAC-3′ and 5′-ACTTGACGTCGTCCCCACCTTCCTC-3′) that were designed based on the 16S rRNA gene sequence of *Nocardia wallacei* ATCC 49872, formerly *Nocardia asteroides* (accession number AY191251), to amplify a PCR product of 1200 bp. The amplicon was purified, and the nucleotide sequence was determined in both directions using the same primers with Taq FS Dye Terminator Cycle Sequencing Fluorescence-Based Sequencing and analysed on an Applied Biosystems 3730 xl DNA sequencing system (Thermo Fisher Scientific, Asheville, NC, USA). The sequence was edited using the Vector NTI 11.5 programme, and a homology search was performed using the GenBank database (nucleotide BLAST 2-16.0 version). The sequence showed 100% identity with *Nocardia brasiliensis* ATCC 706358, *N*. *brasiliensis* 179-09, and *N*. *brasiliensis* DSM 43758 strains, among others. The complete sequence of the 16S ribosomal rDNA gene obtained for *Nocardia brasiliensis* Gea07 strain has been deposited in GenBank under the accession number PV022140.

No primary infectious lesions were detected in the internal organs during extensive examinations (radiography, followed by thoracic computed tomography). The integration of the patient’s clinical case led to the diagnosis of primary cutaneous nocardiosis. The patient was initially treated empirically with trimethoprim/sulfamethoxazole (160/800 mg twice per day); however, after three days of treatment, a non-rash had started to develop on her arms and legs and spread to the chest and axilla. We indicate that the patient come back to the hospital to switch the antibiotic therapy. The antimicrobial treatment was changed to oral amoxicillin–clavulanic acid (875/125 mg every 12 h) monotherapy. Improvement was observed after one month, and complete resolution of the skin lesions was achieved after three months of the same treatment (875/125 mg twice per day). After this time, negative cultures were observed (Figure 1B).

This study was approved by the Institutional Review Board (06-19-10) of the “Dr. Manuel Gea Gonzalez” General Hospital, Mexico City, Mexico.

## 3. Discussion

Nocardiosis is a rare but emerging infectious disease in both immunocompetent and immunocompromised persons. Skin involvement in nocardiosis is classified as follows: (1) primary cutaneous, (2) lymphocutaneous, (3) cutaneous manifestations of disseminated *Nocardia*, and (4) mycetoma. Cutaneous infections may result from primary inoculation or secondary dissemination (particularly immunosuppression).

Primary nocardiosis affects immunocompetent patients with histories of skin trauma. The most common etiological agent associated with primary cutaneous nocardiosis is *N. brasiliensis* [20,21]. Clinical manifestations are divided into lymphocutaneous infection, and acute cutaneous infection (ulcers, abscesses, pustules, cold and warm nodules, granulomas, and localised cellulitis) [11,22]. Lymphocutaneous nocardiosis is defined as the condition where, in addition to skin lesions, there is an inclusion of the regional lymph nodes displaying as nodular lymphangitis. Lymphocutaneous infection begins with ulcerative nodules affecting the regional lymphatic vessels and forming abscesses and may be accompanied by fever [23,24]. This presentation is also frequently called “sporotrichosis type” as the clinical form is indistinguishable from lymphocutaneous syndrome due to *Sporothrix schenckii* infection [25]. Acute cutaneous nocardiosis is a mild form; it occurs one to three weeks after the inoculation and is clinically manifested as cellulitis. Patients with cutaneous lesions should be examined using imaging for lungs and CNS [11,21,26]. Our patient with primary cutaneous nocardiosis had the classic characteristics of superficial skin lesions with lymphocutaneous clinical manifestations characterised by erythematous nodules, with central ulceration in the left thumb, wrist, and inner side of the left forearm and immunocompetent status.

*Nocardia* species have different geographical distributions, virulence patterns, and antimicrobial susceptibility profiles. The classification described by Zhao et al. (2017) [27] separates clinically relevant *Nocardia* species into 13 antimicrobial susceptibility profiles. *N*. *farcinica* may be a more virulent species that is intrinsically resistant to several antibiotics, including third-generation cephalosporins. Therefore, the identification of *Nocardia* strains at the species level is crucial for providing appropriate patient care [20,28,29].

The diagnosis of *Nocardia* infection is usually confirmed by bacterial isolation. Routine examination techniques included direct smear microscopy and culturing. Smear microscopy has poor sensitivity, particularly when the pathogen is present in low numbers in the sample, leading to potential false negatives [17,19]. In addition, the Kinyoun stain demonstrated the acid-fast filaments from the culture and skin lesions. Bacterial cultures showed the highest sensitivity. In Sabouraud dextrose agar, *Nocardia* shows a cotton-like colony, but its growth is very slow, taking 1–2 weeks and sometimes up to 4 weeks. Biochemical techniques are insufficient to accurately discriminate between clinically important species. There are few available biochemical methods compared with the number of potential species, and the interpretation of biochemical test results may require extended incubation times and specific expertise. However, conventional identification methods are currently weak. Molecular studies based on nucleic acid amplification and sequencing have become very useful for precise species identification [17,28,29].

The molecular techniques based on the 16S rRNA gene sequencing are the most widely used and have become the “gold standard” for *Nocardia* species identification [17]. Several genes (*secA1*, *hsp65*, *gyrA*, and *rpoB*) have been evaluated, individually or in combination, to adequately differentiate between *Nocardia* species [17]. In our study presented here, we performed the identification of primary cutaneous nocardiosis using a combination of classical identification by bacterial culture and molecular identification using a PCR amplification and sequencing of the entire 16S rRNA subunit, which allowed species-level identification (*N*. *brasiliensis*) [17,29]. Rapid and specific identification may help clinicians to determine the empirical treatment when an antibiogram is not achieved and enable improved patient management and epidemiological actions.

Matrix-assisted laser desorption/ionisation time-of-flight mass spectrometry (MALDI-TOF) is another method for rapid molecular identification of different microorganisms. This method offers a robust commercial database, low cost, and specific identification at the species level for a wide variety of microorganisms. Ribosomal proteins are the most common molecular markers used to identify microorganisms. Liu et al. (2024) [30] demonstrated that this technique was able to identify 98.7% of 76 *Nocardia* isolates at the species level [30].

Differential diagnosis of other skin infections, such as sporotrichosis, atypical mycobacterial infection, pyoderma, actinomycosis, and botryomycosis, is very important because accurate identification of the causative agent is crucial for establishing a specific antibiotic treatment regimen [31].

Currently, the Clinical and Laboratory Standards Institute (CLSI) broth dilution method is the principal method for performing antimicrobial susceptibility tests (ASTs) on *Nocardia* strains. First-line antimicrobials used for ASTs include amikacin, amoxicillin–clavulanic acid, ceftriaxone, ciprofloxacin, clarithromycin, imipenem, linezolid, minocycline, moxifloxacin, trimethoprim–sulfamethoxazole, and tobramycin. The second-line drugs are cefepime, cefotaxime, and doxycycline [17,32].

To this day, sulfamethoxazole–trimethoprim as monotherapy is the first-line treatment method for all types of nocardiosis. However, with increasing sulphonamide resistance reported amongst diverse species or strains [9,17], other drugs such as carbapenems (imipenem), third-generation cephalosporins (ceftriaxone or cefotaxim), fluoroquinolones (levofloxacin or moxifloxacin), aminoglycosides (amikacin), tetracyclines (minocycline or tigecycline), amoxicillin and clavulanate potassium, and oxazolidinone linezolid are effective alternatives alone or in combination with sulphonamide drugs. Among them, ceftriaxone and imipenem are considered the second-line treatment alternatives. Other options for effective treatment of acute primary cutaneous nocardiosis are amoxicillin–clavulanic acid and intravenous ceftriaxone [33].

Generally, treatment for primary cutaneous forms of the disease should be given for 1 to 3 months. This time is recommended to achieve a complete cure and avoid recurrence in immunocompetent individuals who present skin infections. The duration of treatment is variable and depends on the clinical condition, drug sensitivity of the patient, the severity of the disease, and the immune status of the host [34]. Depending on the site of presentation and evolution of the disease, some patients may require surgery as a complementary treatment [27,31,35]. Sometimes, the disease tends to aggravate if drug levels are inadequate [9].

Sulphonamides are known to commonly cause adverse effects, including allergic reactions, as was the case with our patient; therefore, we chose a high dosage of amoxicillin–clavulanic acid as monotherapy, which resulted in a complete remission, as previously described in other cases [36]. In this sense, amoxicillin–clavulanic acid has been generally used at high doses as an effective treatment for cutaneous nocardiosis: 875/125 mg three times per day (oral), 1.2 g twice per day (intravenous), or 1 g twice per day (oral). In the current case, we used an oral dose of 875/125 mg twice a day, which resulted in a complete resolution of the skin lesions [37,38,39].

Our work has a significant limitation: antibiotic susceptibility testing was not performed. The patient’s immunocompetent condition and allergy to trimethoprim and sulfamethoxazole, the rapid molecular identification of the causative agent (*Nocardia brasiliensis*), and the clinicians’ extensive experience allowed for a rapid response to select the most appropriate and affordable treatment for acute primary cutaneous nocardiosis.

## 4. Conclusions

Nocardiosis is an infection that usually affects immunosuppressed patients. However, primary cutaneous forms may affect immunocompetent patients with a history of trauma presenting with a clinical picture similar to the one described above, and their course and prognosis are usually favourable. Differential diagnoses should include sporotrichosis, atypical mycobacteriosis, and soft tissue infections to avoid incorrect diagnosis and treatment. Nocardiosis may be considered a major mimic of several cutaneous diseases that present with a difficult and often delayed diagnosis.

Early diagnosis using molecular identification, reasonable surgical intervention, and adequate duration of treatment with effective antibiotics are critical for treating these patients. Molecular identification of *Nocardia* infections is mandatory as it allows us to make the correct diagnosis and successfully establish the most effective treatment.

## Figures and Tables

**Figure 1 microorganisms-13-01022-f001:**
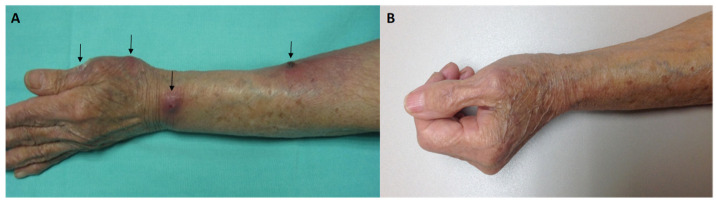
(**A**) Disseminated unilateral four erythematous nodules, with central ulceration, covered by meliceric crusts dermatosis (black arrows). (**B**) Complete remission of disease.

**Figure 2 microorganisms-13-01022-f002:**
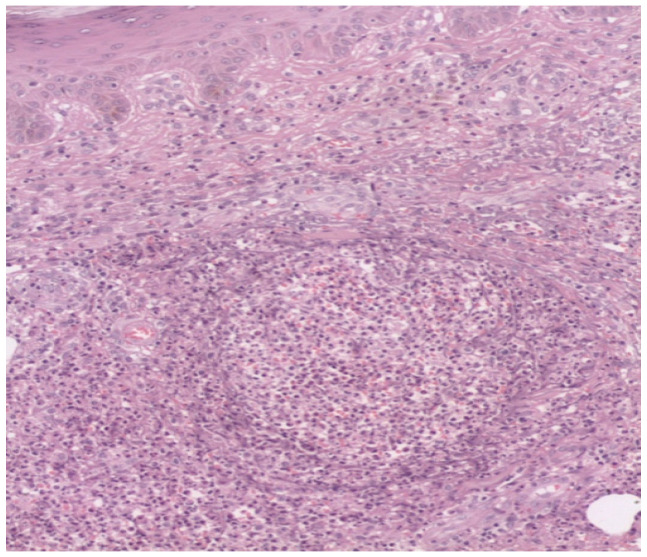
Histologic section showing suppurative granulomatous dermatitis (20×).

## Data Availability

The original contributions presented in this study are included in the article/Appendix A. Further inquiries can be directed to the corresponding authors.

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
