# Peer review of "Primary Cutaneous Nocardiosis (Lymphangitic Type) in an Immunocompetent Patient: A Case Report"

_microorganisms, 2025, doi:10.3390/microorganisms13051022_

Round 1
Reviewer 1 Report
Comments and Suggestions for Authors
(1)The current preface is excessively segmented into multiple paragraphs, resulting in a somewhat disorganized presentation. It is recommended that the initial paragraph of the preface should provide an introduction to Nocardiosis, while the subsequent paragraph should focus on the circumstances of Nocardia spp., the bacteria responsible for causing Nocardiosis.
(2)What is the significance of reporting this case? What makes this case unique? What insights does the reporting of this case offer for the treatment of Nocardiosis? These details should be provided in the introduction section.
(3)Figures 1 and 3 should be juxtaposed for comparative analysis to examine the differences before and after treatment.
(4)It is recommended to provide detailed information on the treatment plan for this case.
(5)In the discussion section, the clinical significance of this case should be addressed. Does the case offer new insights into detection or treatment?
Comments on the Quality of English LanguageThe English could be improved to more clearly express the research.
Author Response
Dear Editor
Above you will find the answer to each of the comments and the suggestions that were kindly given by theby the reviewers to manuscript: Microorganism-3559501 “Primary cutaneous Nocardiosis (lymphangitic type) in immunocompetent patient: A case report”
In the reviewed manuscript we have included each amendment, which are in Word Track Changes along the manuscript. In the following lines, you will find each of the reviewers’ questions appropriately addressed.
Sincerely,
Prof. Carmen Rodriguez-Cerdeira
Corresponding Author
Reviewer 1
1)The current preface is excessively segmented into multiple paragraphs, resulting in a somewhat disorganized presentation. It is recommended that the initial paragraph of the preface should provide an introduction to Nocardiosis, while the subsequent paragraph should focus on the circumstances of Nocardia spp., the bacteria responsible for causing Nocardiosis.
Answer: Lines 46-98: The introduction was re-structured as review suggestion, references were re-assigned.
2)What is the significance of reporting this case? What makes this case unique? What insights does the reporting of this case offer for the treatment of Nocardiosis? These details should be provided in the introduction section.
Answer: Lines 100-107: The significance and details were included inside of the objective.
3)Figures 1 and 3 should be juxtaposed for comparative analysis to examine the differences before and after treatment.
Answer: Line 116: The figures were juxtaposed as figure 1A and B.
4)It is recommended to provide detailed information on the treatment plan for this case.
Answer: Lines 157-165: The details of treatment information was included.
5)In the discussion section, the clinical significance of this case should be addressed. Does the
case offer new insights into detection or treatment?
Answer: Lines 269-278: The clinical significance of this case was included.

Reviewer 2 Report
Comments and Suggestions for Authors
This case report could be more helpful to the reader if it were more about cutaneous inoculation nocardiosis, not about all manifestations of nocardiosis.Cutaneous inoculation nocardiosis with lymphangitis spread, like the reported case, rarely disseminates to other organs and is not treated with intravenous drugs, such as amikacin. By including discussion of pulmonary and disseminated infection, the manuscript is not only much longer but also may confuse the reader into thinking this infection may disseminate and might benefit by intravenous therapy. The figures are good but arrows pointing to the skin lesions would be helpfu.
Author Response
Reviewer 2
This case report could be more helpful to the reader if it were more about cutaneous inoculation nocardiosis, not about all manifestations of nocardiosis. Cutaneous inoculation nocardiosis with lymphangitis spread, like the reported case, rarely disseminates to other organs and is not treated with intravenous drugs, such as amikacin. By including discussion of pulmonary and disseminated infection, the manuscript is not only much longer but also may confuse the reader into thinking this infection may disseminate and might benefit by intravenous therapy.
Answer: We didn't address pulmonary and disseminated infection in the discussion; perhaps it's only mentioned in the introduction. Regarding intravenous therapy, we've included a few lines at the suggestion of reviewer 3, who included this reference as evidence of intravenous therapy (Xu X, Liu Z, Xia X. Acute Primary Cutaneous Nocardiosis. Am J Trop Med Hyg. 2024 26;110:848-849). Our patient was treated with oral therapy. The discussion was improved.
The figures are good but arrows pointing to the skin lesions would be helpful.
Answer: The arrows were included in the figure and yuxtaposed as suggestion of reviewer 2.

Reviewer 3 Report
Comments and Suggestions for Authors
I have carefully studied the manuscript entitled "Primary cutaneous Nocardiosis (lymphangitic type) in immunocompetent patient by Aguilar-Molina H. et al.
The manuscript describes an interesting case of cutaneous infection due to Nocardia in an elderly, female patient which is considered as immunocompetent.
The language used is of average level, lacking serious typos and syntax errors. However, there are some concerns regarding the overall quality of the manuscript; therefore, the authors are kindly suggested to discuss / assess the following issues.
Major issues
1) The authors are kindly requested to follow the CARE guidelines and submit the CARE checklist (https://www.care-statement.org/checklist) as a supplementary file.
2) Crucial information regarding the clinical course of the patient are lacking. The authors are kindly suggested to report all relevant laboratory examinations, especially focusing on those that confirm the patients' immunocompetence.
3) Given that sensitivity tests were not performed, while the patient responded to the administration of amoxycillin / clavulanic acid, the authors are kindly invited to further discuss the key issue of the optimal treatment of cuteneous nocardiosis (for further evidence please see: Xu X, Liu Z, Xia X. Acute Primary Cutaneous Nocardiosis. Am J Trop Med Hyg. 2024 26;110:848-849. doi: 10.4269/ajtmh.23-0584. PMID: 38531103; PMCID: PMC11066350).
Minor issues
Line 63: Please amend "cases4" for "cases".
Lines 257-355: Please report all references in the preferred format.
Author Response
Reviewer 3
Major issues
1) The authors are kindly requested to follow the CARE guidelines and submit the CARE checklist (https://www.care-statement.org/checklist) as a supplementary file.
Answer: The CARE guidelines was included as a supplementary file
2) Crucial information regarding the clinical course of the patient are lacking. The authors are kindly suggested to report all relevant laboratory examinations, especially focusing on those that confirm the patients' immunocompetence.
Answer: Lines 123-126: Laboratory finding was included. We also improve therapy details.
3) Given that sensitivity tests were not performed, while the patient responded to the administration of amoxycillin/clavulanic acid, the authors are kindly invited to further discuss the key issue of the optimal treatment of cuteneous nocardiosis (for further evidence please see: Xu X, Liu Z, Xia X. Acute Primary Cutaneous Nocardiosis. Am J Trop Med Hyg. 2024 26;110:848-849. doi: 10.4269/ajtmh.23-0584. PMID: 38531103; PMCID: PMC11066350).
Answer: Lines 240-249, 259-266: The discussion about therapy and antimicrobial susceptibility was improved.
Minor issues
Line 63: Please amend "cases4" for "cases".
Answer: Line 54: The re-structured introduction moves the original line to 54.
Lines 257-355: Please report all references in the preferred format.
Answer: The references were re-structured and changed according to Microorganism guidelines.

Round 2
Reviewer 1 Report
Comments and Suggestions for Authors
None
Reviewer 3 Report
Comments and Suggestions for Authors
I have carefully studied the revised version of the menuscript entitled "Primary cutaneous Nocardiosis (lymphangitic type) in immunocompetent patient: A case report" by Aguilar-Molina et al.
The authors have adequately responded to all queries raised duting the review process. The quality of the manuscript has been substantially ameliorated. There are no additional comments/issues.